# Evaluation and Analysis of Design Elements for Sustainable Renewal of Urban Vulnerable Spaces

**DOI:** 10.3390/ijerph192416562

**Published:** 2022-12-09

**Authors:** Changzheng Gao, Juepin Hou, Yanchen Ma, Jianxin Yang

**Affiliations:** 1School of Public Administration, China University of Geosciences, Wuhan 430074, China; 2School of Architecture, North China University of Water Resources and Electric Power, Zhengzhou 450046, China

**Keywords:** sustainable development, urban renewal, vulnerable space, grey relation analysis, design elements

## Abstract

The sustainable renewal design of urban vulnerable spaces is critical for urban space quality improvement. Taking Zhengzhou and surrounding cities as examples, a cognitive framework of urban vulnerable spaces is constructed. The three types of urban vulnerable spaces are vulnerable population, vulnerable cultural, and vulnerable forgotten spaces. Their sustainable renewal design elements comprise multidimensional factors, such as functional requirement, space organization, activity facility, urban context continuation, and material texture. The design elements for the sustainable update of urban vulnerable spaces are evaluated by grey relation analysis (GRA), and update strategies are proposed. The result shows that (1) vulnerable population spaces were shown to have the highest sensitivity to functional requirements and activity facility design elements, while vulnerable cultural spaces have high relevance to urban context continuation and functional requirement design elements. Furthermore, space organization, activity facility, and urban context continuation design elements all show high relevance and importance in vulnerable forgotten spaces. (2) The update of vulnerable population spaces should be designed to achieve functional communion; vulnerable cultural spaces can be reshaped through urban context implantation, and vulnerable forgotten spaces can use space creation to enhance ecological space continuity, achieving sustainable renewal. The study provides a reference for decision-making for improving urban vulnerable habitats and the sustainable renewal design of atypical urban space types.

## 1. Introduction

An urban vulnerable space in a city is where the functional requirement is regressive, the form is neglected, and the place is forgotten by people [1]. The sustainable renewal of vulnerable urban spaces is crucial to improve the quality of urban space. The proper handling of issues such as spatial layout, infrastructure, history, and culture determines the potential of sustainable urban development. In addition, urban renewal needs a more refined evaluation model to explore strategies for vulnerable space renewal. In the 1860s, Doxiadis founded the science of human settlement, with urban planning and architecture as the core, and introduced social, cultural, ecological, and technological theories to study the improvement of urban and rural habitats [2]. Since the 1990s, the research on urban renewal has focused on three areas.

The first area is urban infrastructure renewal and overall environmental enhancement. Wu [3] reported that urban renewal requires a comprehensive, detailed urban design and the protection of the city’s cultural characteristics, architecture, and overall environment. Similarly, Vitkova and Silaci [4] noted that urban spatial development should have new functions, structures, and industries, while Li and Y [5] discussed the humanistic urban design of “ecology-landscape-humanity”. In other studies, Tiwari et al. [6] proposed prioritizing sustainable development-oriented urban infrastructure and programs, and Pan and Du [7] suggested improving urban infrastructure, optimizing sanitation conditions, beautifying building facades, and enhancing the participation of residents. These goals achieve the continuous improvement of urban infrastructure. 

Second, the design and construction of urban vitality is another area of focus. For instance, Jane Jacobs’s criteria aim at shaping urban vitality, prioritizing the vibrant urban environment and interpersonal interaction [8]. Wang [9] discussed how to improve people’s perceptions and experiences by creating exquisite and livable environments. In addition, Bai and T [10] discussed how to activate the vitality of urban dead corners through architectural and environmental design and improve the community environment and operations. In a relevant work, Zhuo et al. [11] quantitatively researched urban vitality temporal characteristics, spatial characteristics, coupling types, and dominant modes, and presented the methods to improve urban spatial vitality. 

Finally, constituent elements and inner mechanisms of urban renewal comprise the third focus. For example, Awad and Jung [12], through the planning elements of sustainable urban renewal, concluded that public transportation orientation, restoration of historical and cultural resources, and improvement of energy efficiency tend to be the primary elements. Xu et al. [13] discussed the aesthetic logic of the planes, facades, and material elements of space in the streets and lanes of Nottingham’s historical areas. Furthermore, Yıldız et al. [14] explored the accessibility to social life, resource conservation, quality of the built environment, the protection of vulnerable groups, and business and economic opportunities. In similar research, Zhu and B [15] noted that urban renewal should formulate a design adaptable and tolerant to future changes and proposed the concept of the “urban vulnerable space”. 

In general, urban renewal is more concerned with the renewal of urban physical space and the exploration of specific architectural design cases. Although some scholars have explored the components and formation mechanisms of sustainable urban space renewal, they mostly focused on the macro level of urban and rural planning. Little attention has been paid to the design elements and inner roots of urban vulnerable space renewal. Additionally, the research methods mostly adopt the design summaries of cases, lacking the quantitative evaluation of sustainable renewal design elements. Moreover, the research on the quantitative evaluation of urban vulnerable spaces using interdisciplinary methods is even weaker.

Grey relation analysis (GRA) is an analysis method in grey system theory which uses grey correlation degree values to describe the strength of the relationship between the elements and measure the degree of correlation between system elements [16]. GRA has been applied increasingly as a research method in management, economics, sociology, and other disciplines. One application is urban and rural planning, which involves sustainable urban development, regional research, transportation systems, and industrial development. Weighted analyses of the sustainable development levels of Chinese cities used GRA [17] to study the following: the symbiotic development mechanism of the characteristic elements of traditional village cultural resources in the county [18], the key factors affecting construction projects in Pakistan [19], the sensitivity in the ranking of urban transportation projects in Luxembourg [20], and the urban poverty index in Sabzawa, Iran [21]. 

GRA is mostly used in data processing and ranking research, and few scholars have applied it to the level of architectural space analysis. The design elements of urban vulnerable space renewal have the problem of few data and small samples due to the diverse site environment and changing functional needs. Therefore, by establishing the cognitive framework of urban vulnerable space, this paper takes the design cases of Zhengzhou and surrounding cities to extract design elements and use the GRA method to establish an evaluation model to calculate the correlation between updated design elements and urban development needs. The results can provide new ideas for integrating urban vulnerable space renewal design elements with sustainable urban development.

## 2. Theoretical Mechanism Analysis

According to the analyses of the users and the spatial characteristics of urban vulnerable spaces, the main reasons for the formation of the vulnerable space problem can be summarized using terms such as vulnerable crowds, a fading cultural atmosphere, and the forgotten space function. To solve the problem, vulnerable spaces are divided into three types—vulnerable population space, vulnerable cultural space, and vulnerable forgotten space—which have different spatial characteristics. The design elements of sustainable renewal are comprised of multidimensional factors, such as functional requirements, spatial organization, activity facility, urban context continuation, and material texture. Combining these elemental indicators through the analysis of design cases, we can (1) extract the index of sustainable renewal elements, (2) rank and score the indicators according to the level of demand, and (3) establish the GRA evaluation model to obtain the correlation degree between the sequence of each type of vulnerable space renewal element and the sum sequence of vulnerable space renewal elements. Based on the calculation results, the design paths of sustainable regeneration, such as functional integration, urban context implantation, and spatial creation, are discussed. Figure 1 shows the cognitive framework of the urban vulnerable space.

### 2.1. Root Cause of Vulnerability Problems

The root causes of urban vulnerable space problems mainly include population vulnerability, a decline in culture, and functional forgetting. Population vulnerability is present when, despite many vulnerable populations using the urban activity space, security is lacking, the barrier-free design is not ideal, infrastructure is lacking, and the physical and psychological needs of the vulnerable population are not being met. The decline in culture is reflected in the weakening of the cultural atmosphere and sense of place. In addition, the people’s sense of identification with traditional culture is reduced, their cultural inheritance is gradually squeezed by urban development and new buildings, and the unique urban form is destroyed [22]. Functional forgetting is defined as the existence of empty corners, unused alleyways, and abandoned structures in the city’s busy areas with many high-rise buildings, resulting in the inadequate use of the site and forgotten space or function.

### 2.2. Vulnerable Space Types

Based on root cause analysis, the urban vulnerable space can be divided into the vulnerable population, vulnerable cultural, and vulnerable forgotten space types. Vulnerable population space refers to a place used by vulnerable populations that is not adequately safe or suitable, such as an activity space for elderly people and children or a resting space along a street. A vulnerable cultural space has important historical significance or local culture, where the cultural atmosphere is gradually fading and significant local characteristics are rapidly changing and declining. In this type of space, blurred boundaries develop along with incomplete morphology in areas around memorial sites and historical blocks. Vulnerable forgotten spaces refer to spaces with a single function, poor convenience, and “deleted memory” that prevent them from performing their assigned roles. They experience traffic interference, garbage dumping, noise invasion, poor management, and other hazards and include bus stops, performance venues, and business centers, among others.

### 2.3. Vulnerable Spatial Characteristics

Vulnerable population spaces are characterized by a lack of space for the elderly and young persons to share activities. In addition, the activity venues designed for them ignore the interaction and care needs of the older persons and the behavior needs of the children. These sites are mostly simple combinations of hard pavement, fitness equipment, resting areas, and green vegetation, with less selective activity content or space for safe, comfortable, and interactive behaviors. 

In a vulnerable cultural space, because of the abstraction and complexity of culture, urban design is difficult to guide intuitively and systematically. Large-scale renewal has separated the continuum of urban history. Driven by economic interests, the space environment of architectural monomers and blocks has been destroyed, damaging the aesthetic value, cultural memory, and heritage continuity value of historical blocks. 

Vulnerable forgotten spaces are often small, unclean, and densely populated, serving a mixed population and gradually hiding their functions. They appear in important public places or major node spaces in cities. Because of the high-density development in cities, the original activity spaces around buildings are often surrounded by parking or inapplicable architectural facilities, interrupting the continuity of the block, isolating each activity space, and causing spatial sequence chaos. Some small, seemingly pleasant spaces are eventually ignored, obscured by huge billboards and tall, new buildings.

## 3. Data and Methods

### 3.1. Research Scope and Objects

The research object is selected from six design cases for the renewal of vulnerable spaces in Zhengzhou and surrounding cities in China. These cases address the common and individual issues of different types of vulnerable spaces, involving street rest space, large commercial space governance, small service space, old building renovation, and other aspects. The design elements used in these cases are typical of the multidimensional components of sustainable urban space renewal. The cases are described as follows and shown in Figure 2: (1)On-street charging stations function as rest places, publicity locations, automatic sales, and electric vehicle charging stations. This type of station is a small, leisure landscape construction. The design concept involves the rotation of a rectangular wall and golden-ratio cutting, a modular assembly production process, and a photovoltaic board. A microalgae bioelectric power generation device supplies light. The utilization rate of the population is improved by integrating design elements; hence, it is classified as a vulnerable population space [23].(2)A riverside landscape station is a waterfront strip-shaped leisure park. The surrounding functions are mainly residence, commerce, and education. The site has a good view of landscape but weak water-friendly space. The landscape station consists of two parts, a semi-indoor space and a water-friendly space, which functions as a rest spot and restroom. The design is inspired by the ripples on the water’s surface. The facade skin is a diamond-shaped hollow. The materials are grey tiles and bricks, and this is classified as a vulnerable population space.(3)The Zhengzhou Dehua block renewal belongs to the city’s historic district renovation. Its purposes are to reshape the old street style, increase the rest space, and enhance the shopping experience. The three methods of interface micro-restoration, space micro-intervention, and urban context micro-reconstruction aim to achieve the continuity and integrity of the space, reshaping the monumental atmosphere and urban context. This space is classified as vulnerable cultural space [24].(4)The Kaifeng ancient well pavilion is located in a Jewish settlement. The design includes the ancient well at the center, small open buildings to protect the ancient well, and a restored traditional living space facing the street. Forms including seven-star candlesticks, six stars, and pointed arches microshape the site, and local green bricks are used in the surrounding buildings and vignettes, which are classified as vulnerable cultural spaces [23].(5)A “space under the viaduct” is located under the viaduct of the city’s main road. The surrounding environment for people and vehicles is complex. The design is adapted to the space and site characteristics. The longitudinal space has simple L-shaped concrete modules to create a beautiful and pleasant rest space, classified as a vulnerable forgotten space.(6)The bus stop waiting hall is surrounded by the Zhengzhou Old Cultural and Creative Park, the old factory area, residential communities, and urban business district. The design style conforms to the surrounding cultural and creative park buildings. The materials are profiled steel, concrete, glass, and steel plates, matching the surroundings. Weathering steel plates, linear vertical and horizontal distribution, and circular hole decorations form a contrast in form and material for this vulnerable forgotten space.

### 3.2. Data Collection

By reviewing literature and consulting experts, the sustainability indicators of urban vulnerable spatial renewal elements are extracted from five aspects: functional requirement, space organization, activity facility, urban context continuity, and material texture. The selected cases in this paper cover three types of urban vulnerable spaces that are typical and representative, allowing us to extract universal design techniques. The design elements involved in the above cases are classified according to the degree of demand, forming a table of indicators for urban sustainable renewal elements of urban vulnerable spaces (Table 1).

### 3.3. Evaluation Model Construction

The climate, environment, and cultural characteristics of each city are different, and, thus, the evaluation of urban space is often complicated and difficult to quantify. The methods used analyze and evaluate the core elements of sustainable urban renewal design for vulnerable spaces are key to solving its problems. Through expert interviews, this paper analyzes five key elements: functional requirements, space organization, activity facility, urban context continuation, and material texture, as shown in Figure 3. 

The functional requirement element provides the material basis to meet the physical and psychological needs of local residents. The element of space organization improves the urban space and enhances spatial continuity; it is the noumenon and core of strengthening urban space. Next, activity facility elements achieve humanized settings with applicable and safe characteristics, while the urban context continuation element enhances the unique emotion and sense of belonging within the city by excavating the city’s heritage. Finally, the material texture element is the characteristic of the material that expresses aesthetic feelings. Different types of vulnerable spaces have different demands on design elements. This paper assigns points according to the degree of demand and uses the GRA method to establish the evaluation model of vulnerable space renewal design elements.

Suppose A is the urban vulnerable space, $A_{i}$ is the i-th vulnerable space type, $A_{ij}^{\left(0\right)}$ is the j-th of the sum of the update element of the i-th vulnerable space, $A_{ij}^{\left(1\right)}$ is the sum of all types of vulnerable space update elements, $A_{ij}^{\left(2\right)}$ is the j-th of the functional requirement element of the i-th vulnerable space type, $A_{ij}^{\left(3\right)}$ is the j-th of the space organization element of the i-th vulnerable space type, $A_{ij}^{\left(4\right)}$ is the j-th of the activity facility element of the i-th vulnerable space type, $A_{ij}^{\left(5\right)}$ is the j-th of the urban context continuation element of the i-th vulnerable space type, $A_{ij}^{\left(6\right)}$ is the j-th of the material texture element of the i-th vulnerable space type, and $x_{ij}^{\left(k\right)}$ is the scoring value of $A_{ij}^{\left(k\right)}$ (k=0,1,2,3,4,5,6). In general, $A_{ij}^{\left(k\right)}$ resources are given different scores according to the demand degree (high, medium, and general). Then, the scoring sequence $X^{\left(k\right)}=(x_{1}^{\left(k\right)},x_{2}^{\left(k\right)},\cdots ,x_{n}^{\left(k\right)})$ of the vulnerable space ***A*** is obtained, where $x_{i}^{\left(k\right)}=\sum_{j=1}^{n_{i}^{k}}x_{ij}^{\left(k\right)}$, $x_{i}^{\left(k\right)}$ is the scoring value of the k-type relevant factor of the i-th vulnerable space, and $n_{i}^{k}$ is the number of relevant factors of type *k* in the i-th vulnerable space.

The grey relational coefficient is calculated as follows in Equation (1):(1)$$\xi_{0i}^{\left(k\right)}=\frac{\underset{k}{\min}\;\underset{i}{\min}\left|x_{i}^{\left(k\right)}-x_{i}^{\left(0\right)}\right|+\rho \;\underset{k}{\max}\;\underset{i}{\max}\left|x_{i}^{\left(k\right)}-x_{i}^{\left(0\right)}\right|}{\left|x_{i}^{\left(k\right)}-x_{i}^{\left(0\right)}\right|+\rho \;\underset{i}{\max}\;\underset{k}{\max}\left|x_{i}^{\left(k\right)}-x_{i}^{\left(0\right)}\right|},$$
where $\xi_{0i}^{\left(k\right)}$ is the correlation coefficient of the sum of the vulnerable space update design sequence $X^{\left(0\right)}$ and the k-th element sequence $X^{\left(k\right)}$ in the i-th vulnerable space types and ρ is the resolution coefficient. In general, ρ=0.5, and thus, this value is used for calculations in this paper. In this model, $\underset{k}{\min}\;\underset{i}{\min}\left|x_{i}^{\left(k\right)}-x_{i}^{\left(0\right)}\right|$ and $\underset{k}{\max}\;\underset{i}{\max}\left|x_{i}^{\left(k\right)}-x_{i}^{\left(0\right)}\right|$ are the minimum and maximum differences between the two levels, respectively.

The grey relational degree between the sum of the vulnerable space update element sequence and the category k-type element sequence is defined in Equation (2):(2)$$\gamma (X^{\left(0\right)},X^{\left(k\right)})=\frac{1}{n}\sum_{i=1}^{n}\xi_{0i}^{\left(k\right)}$$

From the definition of GRA, as $\gamma (X^{\left(0\right)},X^{\left(k\right)})$ increases, the connection between the sum of the vulnerable space update element sequences and the k-type element sequence becomes closer. Therefore, $\gamma (X^{\left(0\right)},X^{\left(1\right)})$ can reflect the degree of demand for urban renewal based on the sum of various vulnerable space renewal elements, $\gamma (X^{\left(0\right)},X^{\left(2\right)})$ can reflect the requirement degree of the functional requirement element, $\gamma (X^{\left(0\right)},X^{\left(3\right)})$ can reflect the demand degree of the space organization element, $\gamma (X^{\left(0\right)},X^{\left(4\right)})$ can reflect the demand degree of the activity facility element, $\gamma (X^{\left(0\right)},X^{\left(5\right)})$ can reflect the demand degree of the urban context continuation element, and $\gamma (X^{\left(0\right)},X^{\left(6\right)})$ can reflect the demand degree of the material texture element. Among them, the closer a certain element sequence is to the sum sequence of updated elements, the more important the optimization of a certain element.

## 4. Results

### 4.1. Analysis and Statistics

Based on the above indicators, we apply the GRA method to use various renewal elements for different types of vulnerable spaces as a sequence of relevant factors. The levels of demand are divided into high, medium, and general, which corresponds to a score of 3, 2, and 1, respectively. If no corresponding level of demand in this type of vulnerable space is applicable, the assigned score is zero. The sum of vulnerable space update elements, which is used as a system feature sequence, includes the relevant factor sequences after removing duplicate update elements. Table 2 lists the sequence of vulnerable space sustainable renewal elements and assigned scores.

### 4.2. Evaluation Model Results

We conduct calculations for the grey relational degree between various renewal elements and the sum of the renewal elements of the vulnerable space. The results include the degree of demand for the renewal of the vulnerable space by functional requirement, space organization, activity facility, urban context continuation, and material texture, where $X_{1}(k)$ is the correlation degree between the sum of the updated elements of the vulnerable population space, the vulnerable cultural space, the vulnerable forgotten space, and the sum of the update elements of the vulnerable space. Table 3 shows the correlations of sustainable renewal elements for vulnerable spaces.

A comparison of the correlation degrees shows that each type of vulnerable space has a different degree of sensitivity to each element. Table 3 indicates the following: (1)The sum of renewal elements of each type of vulnerable space is calculated as a sequence of correlation factors. The sum of renewal elements of vulnerable spaces is calculated as a sequence of system characteristics, and the three correlation values obtained are 0.7814, 0.7120, and 0.6416. These results indicate that, compared with the vulnerable cultural space and vulnerable forgotten space, the renewal design of the sustainable renewal of spaces for vulnerable people is representative and comprehensive. Eight indicators have a high-demand for the sum of spatial renewal elements for a vulnerable population space with 24 points. Six indicators have medium-demand with 12 points, and three indicators have general-demand with three points. The sum series of spatial renewal elements for vulnerable people was the most highly correlated with the sum series of spatial renewal elements for vulnerable people.(2)In the vulnerable population space, the correlation degrees between functional requirement and activity facility elements are relatively high at 0.8195 and 0.8143, respectively. This result indicates that functional requirement and activity facility elements are the most important in the renewal design of vulnerable population spaces. For example, using on-street charging stations and the riverside landscape station as data sources, we have three high-demand indicators of functional requirement with nine points, three medium-demand indicators with six points, and two general-demand indicators with two points. Furthermore, we have one high-demand indicator of activity facility with three points, one medium-demand indicator with two points, and zero general-demand indicators with zero points.Among the sequences in the vulnerable population space, the functional requirement sequence has the highest correlation with the system characteristics sequence, and the activity facility sequence exhibits the second highest correlation with the system characteristics sequence. The elements of space organization and material texture are 0.7532 and 0.7658, respectively, indicating that space organization and material texture are closely related to human activities and feelings. As a result, the design should be actively considered and handled properly. The urban context continuation factor is 0.6810, indicating that urban context continuation is relatively insensitive to vulnerable populations, and thus, it can be omitted from the focus of the design.(3)In vulnerable cultural spaces, the urban context continuation and functional requirement are relatively high at 0.8669 and 0.8546, respectively, indicating that they should be the focus of the renewal design of vulnerable cultural spaces. For example, using the renewal of Erqi Dehua District in Zhengzhou and Kaifeng ancient well pavilion as data sources, there are three high-demand indicators of urban context continuation with nine points, three medium-demand indicators with six points, and one general-demand indicator with one point. In addition, there are three indicators having a high-demand for functional requirement with nine points, two medium-demand indicators with four points, and one general-demand indicator with one point.Among the vulnerable cultural spaces, the urban context continuation continuity sequence has the highest correlation with the system characteristic sequence, while the functional requirement sequence has the second highest. The elements of space organization and activity facility are 0.7735 and 0.7696, respectively, reflecting that space organization and activity facility are also relatively important. Thus, the space should be organized with the need for urban context continuation, and the activity facility should be set up with the functional requirement. The material texture is 0.6434, and the impact is relatively small.(4)In the vulnerable forgotten space, space organization, activity facility, and urban context continuation elements are relatively high at 0.7316, 0.7407, and 0.7407, respectively, indicating that they are important in renewal, exhibiting characteristics of mutual influence and mutual promotion. For example, using the space under the viaduct and bus stop waiting hall as data sources, there are two high-demand indicators of spatial organization with six points, one medium-demand indicator with two points, and one general-demand indicator with one point. Additionally, there are two indicators of high-demand for activity facility with six points, two medium-demand indicators with four points, and zero general-demand indicators with zero points. Finally, we have one indicator of high-demand for urban context continuation with three points, one medium-demand indicator with two points, and zero general-demand indicators with zero points.In the vulnerable forgotten space, the spatial organization, activity facility, and urban context continuation sequences are highly correlated with the systematic characteristic sequences. The functional requirement and material texture elements are 0.6434 and 0.6882, respectively, which should be designed based on the spatial organization, activity facility, and urban context continuation elements in the design.

## 5. Discussion

Different types of vulnerable spaces have specific user groups, functions, and spatial feelings. To improve the operability of the renewal design, based on the results of the evaluation model, the design elements with the most significant impact on each type of vulnerable space (the design elements with the highest correlation value) are used as design entry points to achieve the corresponding urban space sustainable regeneration goals. The vulnerable population space renewal continuously updates the urban functional space by means of functional integration, while the vulnerable cultural space renewal remolds the vitality of the urban cultural space by urban context implantation. Additionally, the vulnerable forgotten space renewal through space creation gradually transforms the urban ecological space. The following sub-sections describe our recommendations.

### 5.1. Functional Integration—Continuously Update the Urban Functional Space

To activate the spatial behavior of urban vulnerable population space and improve the quality of life, we can promote the integration of older and younger populations, strengthen the diversity and safety of site functions, and enhance the participation of residents and professionals. In terms of functional diversity, we should set up a complex functional space that integrates the older and younger persons, combines dynamic and static, and meets the needs of all for rest, play, and care. For instance, in the case of on-street charging stations, the functions include rest, toilets, electric vehicle charging, shelter from wind and rain, and automatic sales. Increasing the interaction between people and vegetation landscapes, water features, or rest space improves the openness of the block and provides a variety of daily activities for residents. The water-friendly and landscape viewing functions in the case of a riverside landscape station are examples.

In terms of safety, attention should be given to crowd activities, walking transportation, and social characteristics to help vulnerable populations recognize the environment and feel safe. The activity needs of vulnerable populations are achieved by means of separating people and vehicles, installing clear signs, and improving barrier-free designs. Community participatory design is conducive to more flexible and applicable space, and urban functional space renewal should adhere to co-construction and sharing. Designers should enter the community and let the public and people from various fields contribute ideas. Based on actual community resources, relationship networks and families need to work out a suitable plan for the current situation.

### 5.2. Urban Context Implantation—Remolding the Vitality of Urban Cultural Space

The vulnerable cultural space integrates the emerging urban space with historical buildings by extending the alley emotion and integrating the cultural elements. In terms of continuation of local cultural spirit, the daily life of residents shows the unique emotion to the block. We need to excavate cultural heritage, use local materials, integrate local and ethnic characteristics, protect the authenticity and integrity of buildings, and continue the traditional pattern, historical features, and spatial scale. For example, in the case of the Erqi Dehua District in Zhengzhou, a cultural connection space was established, and the axis was strengthened. Furthermore, in the case of the Kaifeng ancient well pavilion, local bricks were used to express local culture.

In terms of integrating urban context elements, acupuncture-moxibustion implants and repair gradually expand the urban context network so that culture penetrates all aspects of residents’ daily lives. For example, in both cases, the cultural space was repaired and implanted to form cultural interpretation and exhibition spaces. The design of new buildings or the reconstruction of old buildings should absorb and integrate traditional cultural connotations, realize the rebirth of urban cultural space, and achieve the harmonious development of urban locality and modernization.

### 5.3. Space Creation—Gradual Transformation of Urban Ecological Space

Vulnerable forgotten spaces can improve the urban living environment by protecting urban ecological space, strengthening urban spatial sequence, rectifying urban traffic, and increasing usable space. In terms of urban ecological spaces, natural features, such as mountains, rivers, and vegetation should be protected; strengthen the planning and governance of urban nodes, landscape axes, and ecological corridors; enhance the correlation between urban space and the environment; establish ecological infrastructure; integrate urban landscape structure; and improve the overall urban ecological landscape. For example, the case of space under the viaduct forms a good urban ecological space by optimizing the site organization, improving the environment, and increasing green space. The urban space sequence can strengthen the axis processing; use space rhythm, contrast, and other techniques to improve the transition space in the sequence; and increase the continuity and diversity.

In terms of urban transportation, a reasonable organization should focus on the people and vehicles on different roads, landscape points, and squares, with planned guidance and control, to restructure dynamic traffic spaces and static resting places. For example, the bus stop waiting hall case optimizes the relationship between people and vehicles through urban traffic by enlarging the waiting space and creating a good resting landscape. In terms of creating usable spaces, we can organically combine sports activity venues, green parks, and commercial complexes to meet residents’ activity and rest needs at multiple levels, increase the interest in the space, allow for rest space and art installations, stimulate urban creativity, and achieve the gradual transformation goal.

## 6. Conclusions

This paper takes a novel approach by using the GRA method for the quantitative evaluation of urban vulnerable space sustainable renewal design elements. In turn, it proposes paths for the sustainable renewal design of different types of urban vulnerable spaces. We incorporate two aspects:(1)The construction of a cognitive framework of urban vulnerable space is according to the root causes of the vulnerable problems, vulnerable populations, culture fading, and functional forgetting. We divide the urban vulnerable space into three types: vulnerable population space, vulnerable cultural space, and vulnerable forgotten space. Urban vulnerable spatial characteristics, such as a lack of inclusive space for the elderly and young populations, excessive block size, and chaotic spatial sequence, were identified.(2)Five key elements of sustainable renewal of urban vulnerable spaces—functional requirement, space organization, activity facility, urban context continuation, and material texture—were extracted for research. Using GRA, the correlation between the sequence of design elements of each type of vulnerable space and the sequence of the sum of renewal elements of vulnerable space is evaluated. Then, the key elements of renewal design for each type of vulnerable space are clarified. Combined with the key elements proposes the design path and method of urban vulnerable space renewal design via functional integration, urban context implantation, and space creation to achieve the goal of the sustainable renewal of urban functional space, remolding of urban cultural space vitality, and gradual transformation of urban ecological space.

The city is a complex system with rapid development, and the vulnerable space design is only an atypical and “short slab” part of spatial planning and urban renewal which has a profound impact on the enhancement of urban vitality. This paper proposes a sustainable design approach for the renewal of urban vulnerable spaces from a comprehensive and refined analysis of multiple dimensions. The results can provide a reference for decision-making for the improvement of urban vulnerable habitats. In a follow-up study, the grey system theory can be considered to analyze the correlation between the characteristic elements of urban living space and those of an urban landscape system. The goal is to improve the action mechanism of each characteristic element in the city and further expose the law of sustainable urban renewal design.

## Figures and Tables

**Figure 1 ijerph-19-16562-f001:**
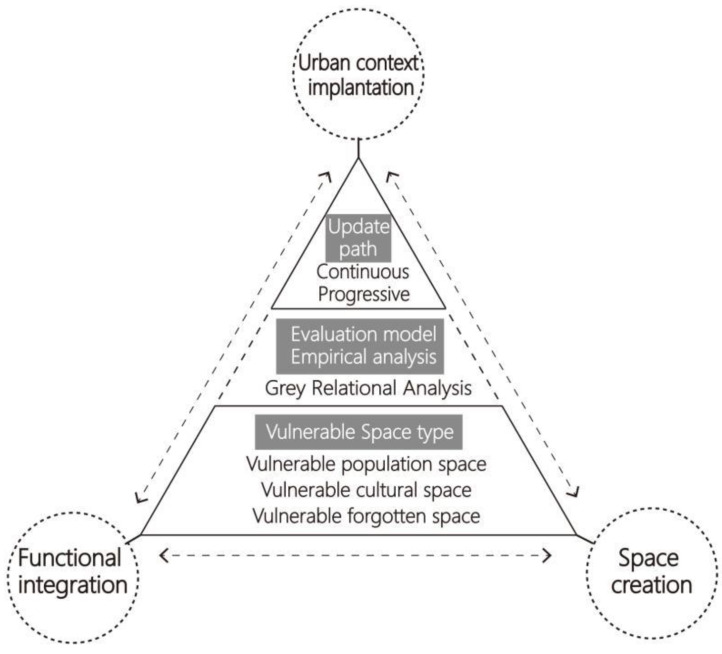
Cognitive framework of urban vulnerable spaces.

**Figure 2 ijerph-19-16562-f002:**
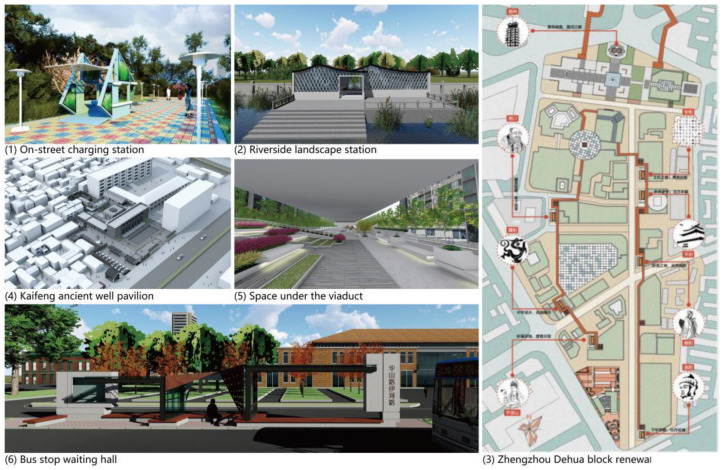
Zhengzhou vulnerable space renewal design scheme.

**Figure 3 ijerph-19-16562-f003:**
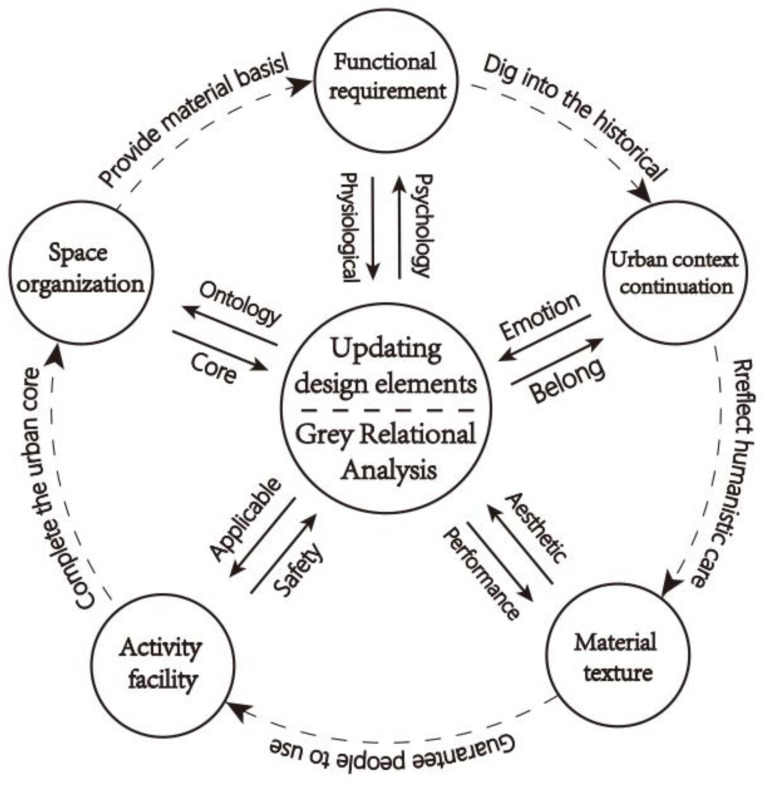
Design elements for sustainable renewal of urban vulnerable spaces.

**Table 1 ijerph-19-16562-t001:** Indicators for urban sustainable renewal elements of urban vulnerable spaces.

		Vulnerable Population Space ^1^	Vulnerable Cultural Space ^2^	Vulnerable Forgotten Space ^3^
Update elements	Functional requirement	High demand—rest, interaction, toilet function; Medium demand—electric vehicle charging, shelter from wind and rain, automatic sales; General demand—water-friendly function, landscape viewing	High demand—cultural expression, commemoration, interaction; Medium demand—cultural activities, cultural business; General demand—rest	High demand—conversation, resting, waiting; Medium demand—shelter from wind and rain
	Space organization	High demand—add rest space, optimize people’s travel paths; Medium demand—provide water-friendly space	High demand—increase the commemorative space, expand the plaza space, establish a continuous interface; Medium demand—elevate the ground floor, provide framed views, increase rest space, add meditation space; General demand—design a building facade	High demand—optimize site organization, remediate the environment; Medium demand—increase green space; General demand—increase waiting space
	Activity facility	High demand—rest facility; Medium demand—water-friendly space	High demand—conservation and viewing facilities, rest facility, leisure sports facility; Medium demand—commercial facility; General demand—residential living space	High demand—rest facility, leisure sports facility; Medium demand—overhead platform, waiting platform
	Urban context continuation	High demand—tiny interventions	High demand—repair the cultural space, create a space for cultural connection, strengthen the axis; Medium demand—provide cultural space implantation, cultural explanation space, cultural exhibition space; General demand—national cultural expression	High demand—interpret the surrounding Urban context; Medium demand—abstract cultural elements
	Material texture	High demand—kind; Medium demand—ecological; General demand—modern	High demand—traditional and modern fusion, traditional, rustic; Medium demand—traditional vs. modern	High demand—comfortable; Medium demand—modern; General demand—technological

Note: ^1^ On-street charging stations, Riverside landscape station. ^2^ The renewal of Zhengzhou Dehua block, Kaifeng ancient well pavilion. ^3^ Space under the viaduct, bus stop waiting hall.

**Table 2 ijerph-19-16562-t002:** Sequence and assignment of elements for sustainable renewal of vulnerable spaces.

Vulnerable Space	Sequence	Update Elements	Assign Points to the Degree of Vulnerable Space Renewal Design Requirements		
			High-Demand	Medium-Demand	General-Demand
	System features	The sum of vulnerable space renewal elements	78	44	9
Vulnerable population space	Relevant factors	The sum of vulnerable population space renewal elements	24	12	3
	Relevant factors	Functional requirement	9	6	2
		Space organization	6	2	0
		Activity facility	3	2	0
		Urban context continuation	3	0	0
		Material texture	3	2	1
Vulnerable cultural space	Relevant factors	The sum of vulnerable cultural space renewal elements	45	22	4
	Relevant factors	Functional requirement	9	4	1
		Space organization	9	8	1
		Activity facility	9	2	1
		Urban context continuation	9	6	1
		Material texture	9	2	0
Vulnerable forgotten space	Relevant factors	The sum of vulnerable forgotten space renewal elements	27	12	2
	Relevant factors	Functional requirement	9	2	0
		Space organization	6	2	1
		Activity facility	6	4	0
		Urban context continuation	3	2	0
		Material texture	3	2	1

Note: Each update element is calculated only once for each different vulnerable space.

**Table 3 ijerph-19-16562-t003:** Vulnerable space sustainable renewal element relevance.

**Grey** **Correlation Degree**	$X_{1}(k)$ (Type Sum)	$X_{2}(k)$ (Functional Requirement)	$X_{3}(k)$ (Space Organization)	$X_{4}(k)$ (Activity Facility)	$X_{5}(k)$ (Urban Context Continuation)	$X_{6}(k)$ (Material Texture)
Vulnerable population space	0.7814	0.8195	0.7532	0.8143	0.6810	0.7658
Vulnerable cultural space	0.7120	0.8546	0.7735	0.7696	0.8669	0.6434
Vulnerable forgotten space	0.6416	0.6434	0.7316	0.7407	0.7407	0.6882

## Data Availability

Not applicable.

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
