# Peer review of "Evaluation and Analysis of Design Elements for Sustainable Renewal of Urban Vulnerable Spaces"

_ijerph, 2022, doi:10.3390/ijerph192416562_

Round 1
Reviewer 1 Report
(1) The title of the paper needs to be revised, it is to general, not only "Study on..".
(2) The language should be polished by native speakers or experienced companies.
(3) Abstract: line14-line 22. This part is more like method or research steps, not resluts. What are the results of the evaluation?
(4) Introduction: line 39-76. It is not proper to just list the previous studies without your own induction and partition. The comparation of evaluation methods are also not mentioned. The authors should fully discuss the previous studies according to your research area.
(5) The authors selected 6 cases of design for the renewal of vulnerable spaces in Zhengzhou City. However, in the results and discussion part, these cases are not mentioned.
Reviewer 2 Report
The article needs a major review. See the comments in the attached archive.

Round 2
Reviewer 2 Report
The article now is ready to be published.